# Physician-Guided Attention Refinement in Chest Computed Tomography Classification under Label Ambiguity

**Yueh-Chun Liu** [1]                    EUGENELIU1998@GAPP.NTHU.EDU.TW
**Chia-Jung Liu** [2]                          M10082100@GMAIL.COM
**Yu-Hsuan Chen** [3,4]                        C2724201@HOTMAIL.COM
**Chang-Wei Wu** [2]                       SINCEADIDAS@HOTMAIL.COM
**Meng-Rui Lee** [2,4]                             LEEMR@NTU.EDU.TW
**Po-Chih Kuo** [1]                          KUOPC@CS.NTHU.EDU.TW

[1] *Department of Computer Science, National Tsing Hua University, Hsinchu, Taiwan*

[2] *Department of Internal Medicine, National Taiwan University, Hsin-Chu branch, Hsin-Chu, Taiwan*

[3] *Department of Critical Care Medicine, Min-Sheng General Hospital, Taoyuan, Taiwan*

[4] *Department of Internal Medicine, National Taiwan University Hospital, Taipei, Taiwan*

## Abstract

Annotation ambiguity due to inter-physician variability remains a major challenge in medical image analysis. In diseases with ambiguous imaging patterns, diagnosis often requires multi-physician consensus, leading to soft and inconsistent labels. We propose a physician-guided human-in-the-loop (HITL) framework that iteratively refines model predictions through expert feedback. Our approach integrates a 3D convolutional neural network (CNN) with an attention mechanism, where physicians review predictions and Grad-CAM attention heatmaps to provide reassessments and feedback for iterative refinement. Experiments on a private multi-center chest computed tomography (CT) dataset of patients with non-tuberculous mycobacterial (NTM) infection demonstrate improved performance, with internal sensitivity increasing from 0.71 to 0.82 under comparable AUROC, and external AUROC improving from 0.72 to 0.80 alongside sensitivity from 0.66 to 0.80. Physician consensus also improves by over 35% in ambiguous cases. These results highlight the value of incorporating structured expert feedback to improve model performance and clinical decision support in uncertain diagnostic scenarios.

**Keywords:** Computed Tomography, Human-in-the-Loop, Expert-Guided Learning, Attention Mechanism

## 1. Introduction

Chest computed tomography (CT) plays a critical role in pulmonary disease diagnosis and is widely used in artificial intelligence (AI)-based clinical applications. However, deep learning models rely heavily on annotation quality (Thanoon et al., 2023; Singh et al., 2020). In certain clinical scenarios, including non-tuberculous mycobacterial (NTM) infection, diagnosis is inherently ambiguous due to inter-physician variability and delayed microbiological confirmation (Griffith et al., 2007). As a result, annotations are often derived from multi-physician consensus and may reflect uncertainty rather than definitive ground truth (Daley et al., 2020; Van Ingen et al., 2018).

Such label ambiguity can degrade model learning and limit generalization across institutions. While attention mechanisms improve interpretability, they are typically learned without explicit clinical supervision (Almahasneh et al., 2025; Liu et al., 2023; Rahman and Marculescu, 2023; Suman et al., 2021). Human-in-the-loop (HITL) approaches incorporate expert feedback, but existing methods mainly focus on label correction or sample selection without jointly refining model attention.

In this work, we propose a physician-guided HITL framework that jointly refines labels and attention through iterative expert feedback. By focusing on low-consensus cases, the model aligns with physician reasoning and improves diagnostic performance and generalization on internal and external datasets.

## 2. Methods

We adopt a hybrid model combining an attention-based 3D convolutional neural network (CNN) (Liu et al., 2025; Zunair et al., 2020; Ji et al., 2012) for CT volumes with a multi-layer perceptron for clinical features to predict NTM infection likelihood. Our key contribution is a **physician-guided human-in-the-loop (HITL) framework** that iteratively refines model predictions and attention maps through expert feedback.

To address label ambiguity, patient cases are divided into consensus and low-consensus groups based on physician agreement. Consensus cases are split into training, validation, and test sets, while low-consensus cases are partitioned into $R+1$ folds and progressively incorporated over $R$ HITL rounds. An initial base model ($r = 0$) is trained using the consensus training set and the first fold (Fold 0) of low-consensus cases with soft labels (Wei et al., 2023; Peterson et al., 2019). Lung segmentation masks serve as initial attention maps to guide the CNN.

The HITL process focuses on low-consensus cases. At each round $r$, the model is updated through four steps:

**(1) Data Collection:** The training data includes the consensus training set and the $r^{th}$ fold of low-consensus cases, each consisting of CT volumes, lung segmentation masks (generated by BCDU-Net (Azad et al., 2019)), and clinical features such as patient demographics and NTM-related microbiological test results.

**(2) Model Inference:** The model from the previous round generates predicted infection likelihoods $p^r$ and Grad-CAM (Selvaraju et al., 2017) attention heatmaps $h^r$.

**(3) Physician Re-examination:** Physicians review predictions and Grad-CAM heatmaps to provide (i) revised assessments $s^r$ and (ii) binary feedback indicators $f^r$ indicating whether the heatmaps are clinically relevant.

**(4) Model Update:** Revised assessments are averaged to form updated soft labels $l^r$. Physician feedback modulates the attention maps, where positive feedback reinforces and negative feedback suppresses highlighted regions. The resulting attention maps $a^r$, combining feedback-modulated heatmaps with lung segmentation masks, guide model refinement.

## 3. Experiments

The HITL process is conducted for $R = 2$ rounds, dividing low-consensus cases into three folds.

**Dataset.** We evaluate on a private multi-center chest CT dataset of patients with NTM infection (Liu et al., 2025), including 413 cases in the internal cohort and 196 cases in the external cohort. Each case is independently annotated by three physicians.

**Model Performance.** On the internal test set, HITL improves performance compared to the base model, achieving higher sensitivity (+15.0%) and F1-score (+10.6%) with comparable AUROC. On the external test set, HITL significantly improves generalization performance, with AUROC, sensitivity, and F1-score increasing by 10.5%, 21.0%, and 7.3%, respectively (Table 1). These results indicate improved robustness and detection of disease cases under distribution shift.

**Physician Interaction Analysis.** To evaluate clinical utility, we conduct a physician questionnaire on low-consensus cases. Attention heatmaps are considered helpful in 31.18% of cases, while physician consensus increases in 35.48% of previously ambiguous cases. Overall, physicians consider the model predictions clinically reasonable, with perceived usefulness increasing across rounds. These findings suggest improved interpretability and diagnostic agreement in uncertain clinical scenarios.

Table 1: Model performance on the internal and external test sets for models trained without (w/o) and with (w/) the proposed HITL process. Values are reported with 95% confidence intervals. **Best** results are shown in bold.

|  | Metrics | Internal Test | External Test |
|---|---|---|---|
| w/o HITL (base model) | AUROC | 0.8783 (0.8142–0.9337) | 0.7219 (0.6439–0.7950) |
|  | Accuracy | 0.8167 (0.7500–0.8833) | 0.7041 (0.6327–0.7653) |
|  | Sensitivity | 0.7143 (0.6429–0.8276) | 0.6628 (0.5732–0.7191) |
|  | Specificity | 0.9062 (0.8065–1.0000) | **0.7364 (0.6415–0.8214)** |
|  | F1-score | 0.7843 (0.7273–0.8627) | 0.6628 (0.5867–0.7356) |
| w/ HITL | AUROC | **0.8795 (0.7464–0.9185)** | **0.7980 (0.7545–0.8672)** |
|  | Accuracy | **0.8833 (0.7667–0.9333)** | **0.7143 (0.6684–0.7449)** |
|  | Sensitivity | **0.8214 (0.6897–0.8966)** | **0.8023 (0.7407–0.8675)** |
|  | Specificity | **0.9375 (0.8387–1.0000)** | 0.6455 (0.5888–0.7105) |
|  | F1-score | **0.8679 (0.7407–0.9259)** | **0.7113 (0.6286–0.7423)** |

## 4. Conclusion

In this work, we propose a physician-guided human-in-the-loop (HITL) framework that integrates expert feedback into the attention mechanism of a 3D CNN for predicting NTM infection likelihood from chest CT scans and clinical data. Experimental results demonstrate improved performance under label ambiguity and enhanced generalization across datasets, indicating improved robustness under data distribution shift. In addition, physician questionnaire results suggest that the proposed approach facilitates consensus in diagnostically uncertain cases, supporting its clinical utility. These findings highlight the value of incorporating structured physician feedback to improve model robustness and clinical decision support in medical imaging.

## Acknowledgments

This work is supported by National Taiwan University Hospital Hsin Chu Branch - National Tsing Hua University Joint Research Program (NTUH HCH-NTHU Joint Research Program, No: 114QF017E1).

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
