# OpenReview forum: "Physician-Guided Attention Refinement in Chest Computed Tomography Classification under Label Ambiguity"
_MIDL.io/2026/Short_Papers — MIDL 2026 - Short Papers Poster_

### Official Review · Reviewer_SQq4 · 2026-05-06
**Review of Physician-Guided Attention refinement in chest CT classification under label ambiguity**

**Rating:** 4
**Confidence:** 4

**Review:**

This is a very interesting approach with good potential, especially for low-data regimes or for conditions with significant ambiguity among experts. I just have a couple of questions

**Summary:**

The authors propose a physician-guided human-in-the-loop framework to develop automated CT interpretation models. The authors’ system is based on a 3D CNN with an attention mechanism that is fine-tuned by Grad-CAM heatmaps + physician review of past predictions.  They show that this approach leads to improvements in non-TB mycobacterial infection diagnoses in internal and external testing.

**Strengths:**

1. Interesting idea and well-executed short study
2. Could be impactful for low data regimes or for conditions where experts often disagree
3. Strong external testing results with marked improvement in AUC

**Weaknesses:**

1)	Could the feedback for heatmaps be made more general than a binary indicator? It might be useful for a physician to specifically point out which region of the heatmap is less useful
2)	Is there a reason to use Grad-CAM maps instead of direct attention maps from the attention mechanism? Grad-CAM and other saliency approaches have well-established limitations
3)	In the base model, did you try using the finalized soft labels after the HITL process for training? I am curious if it is the full HITL process with attention refinement that is the key contributor or is it just improved (less ambiguous) labels

**Justification Of Rating:**

Nice idea and execution of the study - just some justification in design choices needed to make it stronger

---

### Decision · Program_Chairs · 2026-05-08

Accept (Poster)